# The Rare Entity of Basaloid Thymic Carcinoma: A Multicentric Retrospective Analysis from the Italian Collaborative Group for ThYmic MalignanciEs (TYME)

**DOI:** 10.3390/cancers17020239

**Published:** 2025-01-13

**Authors:** Chiara Catania, Sara Manglaviti, Paolo Zucali, Matteo Perrino, Enrico Ruffini, Luca Di Tommaso, Antonio Mazzella, Lorenzo Spaggiari, Angelo Delmonte, Giuseppe Lo Russo, Marina Garassino, Piergiorgio Solli, Giulia Pasello, Lorenzo Rosso, Filippo Lococo, Guido Rindi, Sara Ricciardi, Fernanda Picozzi, Paraskevas Lyberis, Benedetta Tinterri, Laura Pala, Fabio Conforti, Tommaso De Pas

**Affiliations:** 1Medical Oncology Division, Humanitas Gavazzeni, 24125 Bergamo, Italy; chiara.catania@gavazzeni.it (C.C.); benedetta.tinterri@humanitas.it (B.T.); laura.pala@gavazzeni.it (L.P.); fabio.conforti@gavazzeni.it (F.C.); 2Thoracic Oncology Unit, Medical Oncology Department 1, Fondazione IRCCS Istituto Nazionale Tumori, 20133 Milan, Italy; sara.manglaviti@istitutotumori.mi.it (S.M.); giuseppe.lorusso@istitutotumori.mi.it (G.L.R.); 3Department of Oncology IRCCS, Humanitas Research Hospital, 20089 Rozzano, Italy; paolo.zucali@humanitas.it (P.Z.); matteo.perrino@humanitas.it (M.P.); 4Department of Biomedical Sciences, Humanitas University, 20072 Pieve Emanuele, Italy; luca.di_tommaso@hunimed.eu; 5Thoracic Surgery Department, Città della Salute e della Scienza di Torino, Ospedale Molinette, 10126 Turin, Italy; enrico.ruffini@unito.it (E.R.); paraskevas.lyberis@unito.it (P.L.); 6Department of Surgical Sciences, Thoracic Surgery, Università degli Studi di Torino, 10124 Turin, Italy; 7Department of Pathology, Humanitas Research Hospital IRCCS, 20133 Rozzano, Italy; 8Department of Thoracic Surgery, European Institute of Oncology, IRCCS, 20141 Milan, Italy; antonio.mazzella@ieo.it (A.M.); lorenzo.spaggiari@ieo.it (L.S.); 9Department of Thoracic Surgery, Milan University, 20122 Milan, Italy; 10Istituto Romagnolo per lo Studio dei Tumori (IRST) Dino Amadori, IRCCS, 47014 Meldola, Italy; angelo.delmonte@irst.emr.it; 11Thoracic Oncology Program, The University of Chicago Medicine & Biological Sciences, Chicago, IL 60637, USA; 12Thoracic Surgery Division, Policlinico Sant’Orsola, 40138 Bologna, Italy; piergiorgio.solli@istitutotumori.mi.it; 13Medical Oncology 2, Veneto Institute of Oncology, IRCCS, 35128 Padova, Italy; giulia.pasello@iov.veneto.it; 14Department of Surgery, Oncology and Gastroenterology, University of Padova, 35122 Padova, Italy; 15Thoracic Surgery and Lung Transplantation Unit, Fondazione IRCCS Cà Granda Ospedale Maggiore Policlinico, 20122 Milan, Italy; lorenzo.rosso@unimi.it; 16Department of Pathophysiolgy and Rransplantation, Università degli Studi di Milano, 20122 Milan, Italy; 17Thoracic Surgery Unit, Fondazione Policlinico Universitario A. Gemelli IRCCS, 00168 Rome, Italy; filippo.lococo@policlinicogemelli.it; 18Thoracic Oncology Unit, Università Cattolica del Sacro Cuore, 00168 Rome, Italy; 19Anatomic Pathology Unit, Department of Woman and Child Health, Fondazione Policlinico Universitario A. Gemelli IRCCS, 00168 Rome, Italy; guido.rindi@policlinicogemelli.it; 20Section of Anatomic Pathology, Department of Life Sciences and Public Health, Università Cattolica del Sacro Cuore, 00168 Rome, Italy; 21Department of Cardio-Thoracic Surgery, San Camillo Forlanini Hospital, 00152 Rome, Italy; sricciardi2@scamilloforlanini.rm.it; 22Sarcoma and Rare Tumors Unit, Istituto Nazionale Tumori, IRCCS Fondazione G.Pascale, 80131 Naples, Italy; fernanda.picozzi@istitutotumori.na.it

**Keywords:** thymic cancer, basaloid thymic carcinoma, TYME network

## Abstract

Basaloid thymic carcinoma (BTC) is an extremely rare tumor, and very little data are available on its clinical behavior, drug sensibility, and patients’ outcome. We retrospectively collected demographical, clinical, and pathological data of all consecutive patients previously diagnosed with BTC at TYME-referral institutes from 2008 to 2023. Twenty-eight patients with BTC were identified. A total of 22/28 patients were included in this analysis. BTC is generally diagnosed as a localized disease, and no alterations in actionable targets or microsatellite instability were identified. Patients with stage I–III BTC can achieve long-term DFS, and efforts should be made to perform radical surgical resection, combined with perioperative treatment when appropriate. Patients with advanced disease progression have shown a high response rate to systemic treatments, but they have a poor prognosis.

## 1. Introduction

Thymic epithelial tumors (TETs) are a heterogeneous group of rare neoplasms arising from thymic epithelial cells with a complex histopathologic classification.

The World Health Organization (WHO) classifies TETs into two main subgroups: thymomas (TMs) and thymic carcinomas (TCs) [1,2,3]. TMs include five subtypes (A, AB, B1, B2, and B3), which are distinguished by the morphology of the epithelial cells and the percentage of non-neoplastic lymphocytes. The prognosis is best for type A, progressively worse for the other subtypes, and worst for type B3 [4,5,6,7].

Thymic carcinoma (TC) is a rare malignancy accounting for approximately 10–12% of all TETs. It is the most aggressive TET subtype, with a more aggressive behavior and a higher propensity for widespread metastasis [8,9]. Patients with advanced TC are usually treated with systemic therapies, including platinum-based chemotherapy such as cisplatin/cyclophosphamide/doxorubicin or carboplatinum/paclitaxel combinations and anti-angiogenic agents, such as sunitinib and lenvatinib, achieving a 5-year overall survival rate of approximately 30–55% [10,11,12,13,14,15].

In addition, immunotherapy has shown robust clinical activity both as a monotherapy [16,17,18,19] and in combination [20], and it is considered one of the standard treatment options in many international guidelines.

Basaloid thymic carcinoma (BTC) is an extremely rare subtype of TC that is well recognized in the WHO classification. After the identification of squamous cell marker expression in BTC, this tumor became a rare subtype of squamous cell thymic carcinoma (STC), reclassifying STC into keratinizing, non-keratinizing, and basaloid subtypes. Tumors are classified as BTC when more than 50% of the tumor is basaloid. Tumors with 50% or less basaloid component may be included in the diagnosis “with basaloid features”. The differential diagnosis includes poorly differentiated squamous cell carcinoma and metastatic tumor, with CD-117 being a useful marker in differential diagnosis [21] (Figure 1).

Choosing an appropriate treatment strategy for patients with BTC is often challenging because of its extreme rarity and the very limited data available on its clinical behavior, drug sensitivity, and patient outcomes. To our knowledge, only about 30 cases of BTC have been published in the English literature, and most of them are single-case reports [22,23,24,25,26,27,28,29,30].

In order to provide a basis for clinical decision making in this subgroup of patients, the referral centers of the Italian Collaborative Group for ThYmic MalignanciEs (TYME) retrospectively identified patients diagnosed with BTC, reviewed the pathological diagnosis, and analyzed all available clinical data.

The study was approved by the TYME research group, by the Scientific Ethics Committee of the coordinating institute (CE Humanitas Prot. GAV 312/23 on 23 Maj 2023), and by all the other involved centers.

## 2. Material and Methods

We performed a multicenter retrospective observational study of patients diagnosed with TET from 2008 to 2023 within the referral centers of the TYME network.

TYME (ThyYmic MalignanciEs) is a network established in Italy in 2014 with the aim of promoting collaboration among Italian centers with specific multidisciplinary expertise in TETs. Currently, 23 centers are members of the network. As of December 2024, the first 11 of them have participated in the Italian Registry on Thymic Epithelial Tumors (TETs), and all these centers participated in this retrospective analysis.

Medical records were reviewed to extract data from all patients previously diagnosed with BTC. Inclusion criteria included tumor location in the anterior mediastinum, an absence of other primary tumors, and the presence of basaloid morphologic and histochemical features, specifically a histologic diagnosis of thymic carcinoma, positive immunohistochemical staining for CD117, and a basaloid component of more than 50% of the tumor. Tumors with 50% or less basaloid component, so-called “tumors with basaloid features”, were excluded.

All selected cases were reviewed by the center’s referring pathologist and oncologist to confirm the diagnosis of BTC.

Tumors were defined as localized (stage I–III) or metastatic (stage IV) according to 8th TNM.

Descriptive analyses were performed for all BTC-confirmed patients. Disease-free survival (DFS) and overall survival (OS) functions were estimated using the Kaplan–Meier method on the number of patients with available survival data. DFS was calculated from the date of surgery to the date of relapse or death, whichever occurred first. OS was calculated from the date of diagnosis to the date of death or last follow-up. Tumor response was assessed according to the RECIST 1.1 criteria [31].

## 3. Results

Twenty-eight patients with BTC diagnosed between 2008 and 2023 were identified by 11 centers. A total of 22/28 patients were included in this analysis. Six patients were excluded because of concomitant second primary tumor (one patient), insufficient data (one patient), or unconfirmed diagnosis of BTC at pathological review (four patient).

Patient characteristics are summarized in Table 1.

The median age of the patients at the time of diagnosis was 58 years (range: 33–76). A total of 11/22 patients were male.

The majority of patients had early stage disease at the time of diagnosis (stage I: 9 pts, stage II: 6 pts). Three patients were diagnosed with stage III and four patients with stage IV disease.

All 18 patients with localized disease at diagnosis underwent surgery. Three patients received preoperative chemotherapy, while no patient received preoperative radiotherapy. Ten patients also received postoperative radiation.

All six patients with metastatic disease received first-line chemotherapy, and five received additional systemic therapies.

No patient had autoimmune disease.

### 3.1. Patients with Localized Disease

Eighteen patients out of twenty-two were diagnosed with localized disease (TNM stage I: eight patients, stage II: five patients, stage III: three patients, and not available: two patients).

All 18 patients with localized BTC underwent surgery. Three patients received platinum-containing preoperative chemotherapy consisting of a CAP (cisplatin, doxorubicin, and cyclophosphamide) regimen (two patients) and a carboplatin plus paclitaxel combination (one patient).

According to the RECIST 1.1 criteria, two patients (one patient treated with CAP, and one patient treated with CBDCA plus paclitaxel) achieved a partial response (PR) and remained disease-free after surgery at 10+ and 70+ months. The best response of the remaining patient was stable disease.

Ten patients received adjuvant radiotherapy.

At a median follow-up of 46 (1–133) months, six out of eighteen patients relapsed (two locoregional and four distant). Median overall survival and median relapse-free survival were not reached. At 48 months, OS was 77% (95%CI 43–92), and DFS was 63% (95%CI 30–83) (Figure 2).

### 3.2. Patients with Upfront Metastatic Disease

Four out of twenty-two patients were diagnosed with sincronous metastatic lesions, mostly localized to the lungs, liver, and bones. Four out of twenty-two patients were diagnosed with sinusoidal metastatic lesions, mostly localized to the lungs, liver, and bones. Three patients died, and one patient was alive for 24 months. Median OS was 7 months (Figure 2).

### 3.3. Response to Systemic Treatments

Six patients received first-line systemic treatment for metastatic disease, all with platinum-containing chemotherapy: carboplatin plus paclitaxel (five patients) and carboplatin plus etoposide (one patient). Five patients had a partial response, and one patient had a complete response (CR). Five out of six also received a second-line treatment. Two out of five were treated with sunitinib, and one of them achieved a PR lasting 7 months; one patient received a pembrolizumab/lenvatinib combination with a complete response lasting 7+ months.

### 3.4. Molecular Analysys

Next-generation sequencing was performed in three tumor samples, and no KIT mutations, other actionable alterations, or microsatellite instability were identified.

## 4. Discussion

This retrospective analysis, performed by 11 centers of the TYME network, identified 22 patients with BTC diagnosed over a 15-year period, confirmed by a pathological and oncological internal review.

Thymic basaloid carcinoma appeared to be an extremely rare tumor, usually diagnosed between 50 and 70 years of age, with an equal distribution between men and women.

Of particular interest, BTC was diagnosed in most cases as localized WHO stage I–III disease. All patients underwent radical surgery with or without perioperative treatment. With a median follow-up of 46 (1–133) months, 12/18 patients remained disease-free with a 48 mOS and DFS of 77% and 63%, respectively. These data suggest a similar outcome for patients with localized BTC compared to that reported for patients with stage I–III thymic carcinoma [29].

The lack of adequate studies in this patient setting, due to the extreme rarity of this tumor, makes it impossible to make an adequate comparison between our results and those reported in the literature for patients with BTC. In addition to some case reports [22,23,24,25,26,27,28,29], we can only refer to the clinicopathologic study of 12 cases of BTC by Jeffrey G. Brown et al. reported long-term data in eight patients diagnosed with localized/metastatic BTC [30]. In this report, five out of eight patients died of their disease at an average of 34 months from the time of diagnosis.

The small number of patients treated with preoperative chemotherapy does not allow for definitive conclusions to be drawn about its role in determining patient outcomes. In light of the activity demonstrated in the three patients treated in this series, its use can be considered and deemed useful for an adequate multimodal approach. Three patients were treated with induction carboplatin plus paclitaxel or the CAP regimen, achieving a partial response (two patients) or disease stabilization (one patient). The limited sample size and retrospective nature of this analysis do not allow for the evaluation of the efficacy of adjuvant radiotherapy, which was administered to 10/18 patients with no major side effects.

The response to systemic treatments in patients with advanced BTC appears to be higher than expected when compared with the available data on TC. Tumor responses were observed in all six patients treated with a platinum-containing regimen and in one of the two patients treated with sunitinib. Interestingly, one patient treated with a pembrolizumab/lenvatinib combination achieved a complete response lasting 7+ months.

Despite good drug sensitivity, the prognosis of patients diagnosed with synchronous metastases seems to be poor, with an observed mOS of 7 months in the small sample size analyzed. Unfortunately, no KIT mutations, other actionable target alterations, or microsatellite instability were identified to suggest possible targeted approaches.

The limitations of this study are due to its retrospective nature, the absence of a centralized review, and the limited number of cases analyzed.

## 5. Conclusions

Basaloid thymic carcinoma is an extremely rare tumor, usually presenting as localized disease, and no actionable targets have been found in the few cases analyzed by NGS. Patients diagnosed with stage I–III disease can achieve long-term DFS, and efforts should be made to perform radical surgical resection, combined with perioperative treatment whenever appropriate. Patients with advanced disease have a poor prognosis despite a high response rate to systemic treatments. This experience also demonstrates the potential of collaborative networks to contribute to the knowledge of ultra-rare diseases. A major collaborative effort should be undertaken to better understand the preclinical and clinical basis of BTC in order to improve the treatment of this extremely rare tumor.

## Figures and Tables

**Figure 1 cancers-17-00239-f001:**
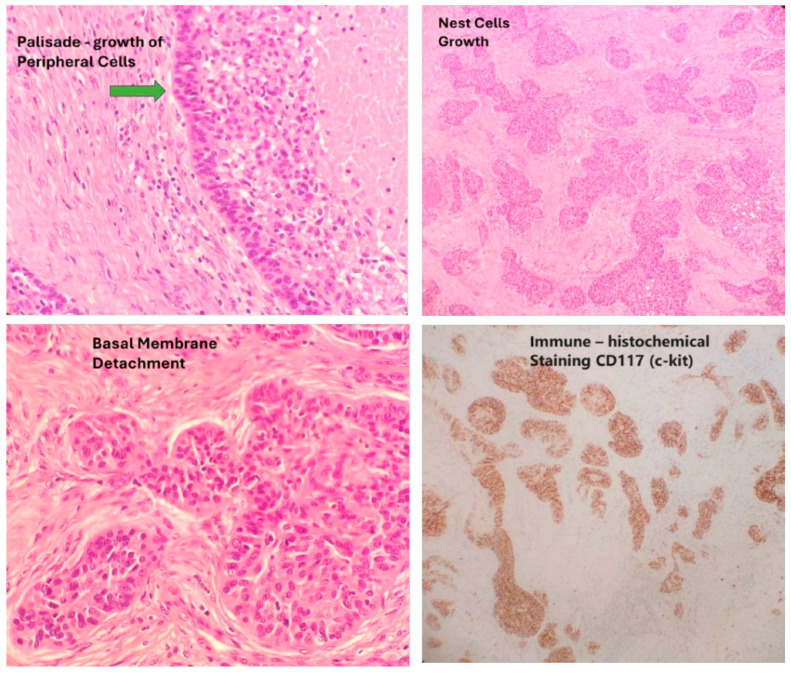
Typical histological features and CD117 staining of BTC (primary tumor specimens).

**Figure 2 cancers-17-00239-f002:**
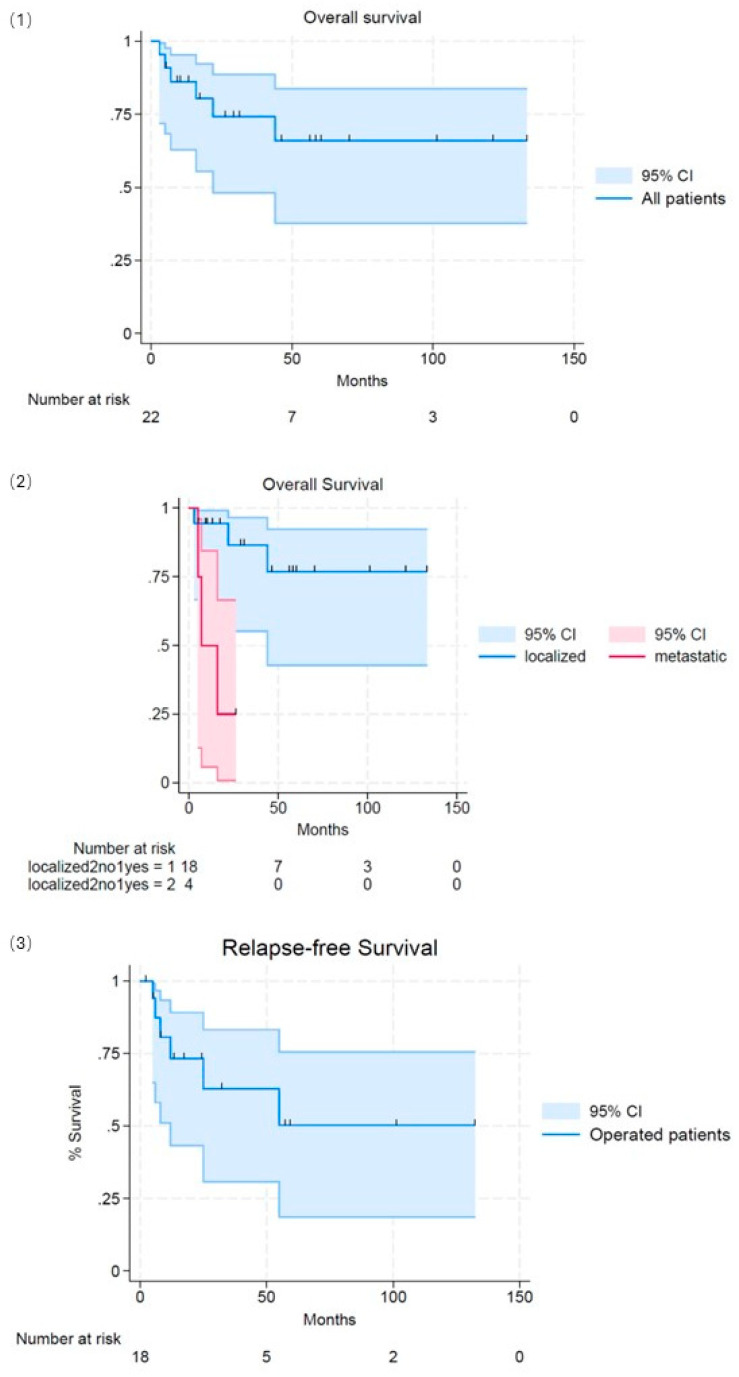
Survival. (**1**): Overall survival of the entire population. (**2**): Overall survival split (localized/metastatic tumor at diagnosis). (**3**): Relapse-free survival split (localized/metastatic tumor at diagnosis).

**Table 1 cancers-17-00239-t001:** Characteristics of patients.

All Patients (*n* = 22).	
Sex	
Male	12 (55%)
Female	10 (45%)
Ethnicity	
White	22 (100%)
Age at diagnosis (years)	
Median	63 (33–76)
Stage at diagnosis	
Localized	18 (82%)
I	9 (41%)
II	6 (27%)
III	3 (14%)
Metastatic	4 (18%)
Treatment of localized disease (18 patients stage I–III)	
Preoperative Radiotherapy	
	None
Preoperative chemotherapy	
Yes	3 (17%)
Cisplatin, doxorubicin, and cyclophosphamide	2
Carboplatin plus paclitaxel	1
No	15 (83%)
Adjuvant radiotherapy	
Yes	10 (56%)
no	8 (44%)
Surgery	
Yes	18 (100%)
No	none
Treatment of stage IV disease (6 patients)	
First line chemotherapy	6 (100%)
Carboplatin plus paclitaxel	5
Carboplatin plus etoposide	1
≥2 lines systemic treatments	5 (83%)
Sunitinib	2
Pembrolizumab plus Lenvatinib	1
Platinum plus etoposide	1
Cisplatin, doxorubicin, and cyclophosphamide	1

## Data Availability

No new data were created or analyzed in this study. Data sharing is not applicable to this article.

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
