# Peer review of "The Rare Entity of Basaloid Thymic Carcinoma: A Multicentric Retrospective Analysis from the Italian Collaborative Group for ThYmic MalignanciEs (TYME)"

_cancers, 2025, doi:10.3390/cancers17020239_

Round 1

Reviewer 1 Report

Comments and Suggestions for Authors

This article demonstrates significant innovation by conducting a retrospective study on a rare type of cancer in clinical practicethymic basaloid carcinomato uncover its disease characteristics and treatment strategies. However, as a manuscript intended to serve as a reference for the field, it requires substantial revisions as outlined below:

1.      To provide a more comprehensive context, the article should include detailed information on the global incidence, mortality rate, and post-diagnosis survival period of thymic basaloid carcinoma.

2.      The clinical and pathological features, as well as specific markers of thymic basaloid carcinoma, should be systematically summarized to enhance the scientific rigor of the study.

3.      If possible, besides provide immunohistochemical and morphological feature images of BTC to further substantiate the diagnostic criteria, the authors could also compare the pathological characteristics and clinical presentations of BTC with those of other thymic carcinoma subtypes.

4.      The basis and criteria used by experts for pathological review and diagnosis, as mentioned in the article, should be explicitly outlined for clarity and reproducibility.

5.      The article currently cites too few references. It would benefit from consulting a broader range of literature and incorporating methods from existing studies to improve the classification and analysis of patient characteristics.

Comments on the Quality of English Language

The English could be improved to more clearly express the research.

Author Response

  1. To provide a more comprehensive context, the article should include detailed information on the global incidence, mortality rate, and post-diagnosis survival period of thymic basaloid carcinoma.

We have added the required data for TET and CT patients in the introduction and also expanded the text adding available data on systemic treatments for CT patients. Regarding data on BTC, unfortunately there are no data in the literature except for a few case reports and one series of 12 patients. In the discussion, we have therefore reported the data of these series and added the references of the case reports.

  1. The clinical and pathological features, as well as specific markers of thymic basaloid carcinoma, should be systematically summarized to enhance the scientific rigor of the study.

It has been done, see introduction

  1. If possible, besides provide immunohistochemical and morphological feature images of BTC to further substantiate the diagnostic criteria, the authors could also compare the pathological characteristics and clinical presentations of BTC with those of other thymic carcinoma subtypes.

It has been done, see introduction and figure 1 with images of histological BTC features

  1. The basis and criteria used by experts for pathological review and diagnosis, as mentioned in the article, should be explicitly outlined for clarity and reproducibility.

It has been done, see matherial & methods

  1. The article currently cites too few references. It would benefit from consulting a broader range of literature and incorporating methods from existing studies to improve the classification and analysis of patient characteristics.

It hase been done, see references (expanded to 31 references). We added in introduction and material & methods classification of TC and BTC and WHO criteria, unfortunately no  more strong data can be added because of the lack of  other studies on BTC.

The English has been improved.

Reviewer 2 Report

Comments and Suggestions for Authors

The authors have reported on the clinical course of BTC in a retrospective study. Their report is valuable and well-compiled. However, several points require revision:

  1. In the abstract and conclusion, phrases such as "To our knowledge, this is the first analysis focused on the outcome of a BTC patient population" are noted multiple times. These statements should be removed entirely. I believe that claiming a study to be the first, or possibly the first, is unrelated to the paper's significance. The importance lies in objectively presenting and reporting the findings. While this is my opinion, excessive emphasis on being "the first" may lower the perceived value of the paper. Additionally, expressions like "To our knowledge" appear in several places, and I request that these be revised. Please consider replacing them with more appropriate expressions.
  2. In the Introduction, include a more detailed explanation of the standard treatments typically performed for BTC.
  3. In the Materials and Methods section, define the terms localized and metastatic.
  4. In Figure 1, correct the "No. at risk" section, which says "localized two no 2 yes =1, 2" to clearly distinguish between localized and metastatic. Additionally, provide figure legends for Figures 1–3.
  5. If you are presenting data separately for localized and metastatic patients in Figure 1, include a table showing the patient background for each group.
  6. It is unclear what chemotherapy regimens were administered for preoperative BTC and Stage IV BTC cases. Please detail the chemotherapy contents in a table format.
  7. There are multiple instances where the abbreviation BTC is incorrectly written as BCT. Please correct these errors.
  8. I did not understand the authors' claim that neoadjuvant therapy is appropriate. In the Results section, describe the clinical course of patients who underwent preoperative chemotherapy. After addressing this, revise the discussion accordingly.
  9. In the Discussion section, include the limitations of this study.

Author Response

  1. In the abstract and conclusion, phrases such as "To our knowledge, this is the first analysis focused on the outcome of a BTC patient population" are noted multiple times. These statements should be removed entirely. I believe that claiming a study to be the first, or possibly the first, is unrelated to the paper's significance. The importance lies in objectively presenting and reporting the findings. While this is my opinion, excessive emphasis an being 'the first" may lower the perceived value of the paper. Additionally, expressions like "to our know/edge" appear in several places, and I request

that these be revised. Please consider replacing them with more appropriate expressions.

Al these statements have been removed

2 In the Introduction, include a more detailed explanation of the standard treatments typically performed for BTC

BTC is an extremely rare tumor, and there is no literature supporting specific treatments for this histotype. Therefore, the typical treatments for TC are necessarily used. These treatments are therefore reported in the introduction, as requested.

Patients with advanced TC are usually treated with systemic therapies, including platinum-based chemotherapy such as cisplatin/cyclophosphamide/doxorubicin or carboplatinum/paclitaxel combinations  and anti-angiogenic agents, such as sunitinib and lenvatinib, achieving a 5-year overall survival of approximately 30 -55% [10-1 5]. In addition, immunotherapy has shown robust clinical activity both as monotherapy [16-19] and in combination [20] and is considered one of the standard treatment options in many international guidelines.

  1. In the Materials and Methods section, define the terms localized and

We added the requested definition in M&M:  Tumors were defined as localized (stage I-III) or metastatic (stage IV) according to 8th TNM

4 In Figure 1, correct the "No. at risk" section, which says "localized two no 2 yes ——1. 2• to cleañy distinguish between localized and metaslatic. Additionally, provide figure Legends far Figures 1—3.

Accepted, Graph 1:

localized 2no 1yes=1 change to “localized

localized 2no 1 yes =2 change to “metastatic”

Figure 1: Legend has been added

5 If you are presenting data separately for localized and metastatic patients in Figure 1, include a

table showing the patient background for each group.

Accepted: We added the data of localized tumors by stage and metastatic tumors in the text and in Table 1 and we report outcome separately in Graph 1

6 It is unclear what chemotherapy regimens were administered for preoperative BTC and Stage IV BTC cases. Please detail the chemotherapy contents in a table format

Accepted All these data heve been added in table 1 and reported in the text

7 There are multiple instances where the abbreviation BTC is incorrectly written as BCT. Please correct these errors

Errors have been corrected

8 I did nOt understand the authors’ claim that neoadjuvant therapy is appropriate. In the Results

section, describe the clinical course of patients who underwent preoperative chemotherapy. Aher addressing this, revise the discussion accordingly.

Accepted We suggest that neoadjuvant chemotherapy is appropriate because It was administered to three patients where the multidisciplinary evaluation deemed this approach useful, and all three had a response or disease control and were able to undergo radical surgery. Naturally, the small sample size does not allow for further observations. We have therefore modified the discussion to specify this concept

The small number of patients treated with preoperative chemotherapy does not allow for definitive conclusions to be drawn about its role in determining patient outcomes. In light of the activity demonstrated in the three patients treated in this series, its use can be considered whenever deemed useful for an adequate multimodal approach. Three patients were treated with induction carboplatin plus paclitaxel or CAP regimen, achieving a partial response (2 pts) or disease stabilization (1 pt).

9 In the Discussion section, include the limitations of this study.

Accepted. We added this statement

The limitations of this study are due to its retrospective nature, the absence of a centralized review, and the limited number of cases analyzed.

Round 2

Reviewer 1 Report

Comments and Suggestions for Authors

I have no other suggestions.

Reviewer 2 Report

Comments and Suggestions for Authors

The authors have responded appropriately to my comments. I have no further remarks.